# Supplementation with Highly Standardized Cranberry Extract Phytosome Achieved the Modulation of Urinary Tract Infection Episodes in Diabetic Postmenopausal Women Taking SGLT-2 Inhibitors: A RCT Study

**DOI:** 10.3390/nu16132113

**Published:** 2024-07-02

**Authors:** Mariangela Rondanelli, Francesca Mansueto, Clara Gasparri, Sebastiano Bruno Solerte, Paola Misiano, Simone Perna

**Affiliations:** 1Department of Public Health, Experimental and Forensic Medicine, University of Pavia, 27100 Pavia, Italy; mariangela.rondanelli@unipv.it; 2Endocrinology and Nutrition Unit, Azienda di Servizi alla Persona ‘‘Istituto Santa Margherita’’, University of Pavia, 27100 Pavia, Italy; francesca.mansueto01@universitadipavia.it; 3Department of Internal Medicine, UOC Geriatrics and Diabetology, University of Pavia, 27100 Pavia, Italy; sebastianobruno.solerte@unipv.it; 4Department of Pharmacological and Biomolecular Sciences, Università Degli Studi di Milano, Via Pascal 36, 20133 Milan, Italy; paola.misiano@guest.unimi.it; 5Division of Human Nutrition, Department of Food, Environmental and Nutritional Sciences (DeFENS), Università Degli Studi di Milano, 20133 Milan, Italy; simoneperna@hotmail.it

**Keywords:** highly standardized cranberry extract, sodium glucose co-transporter 2 (SGLT2) inhibitors, highly standardized cranberry extract phytosome, menopause

## Abstract

Urinary tract infections (UTIs) are the most common bacterial infections in postmenopausal women, and women with diabetes are possibly at a higher risk. The aim of this study is to evaluate the potential benefit on the prevention of UTI episodes, assessed by urinalysis and urine culture (primary outcome) after two, four and six months, of daily oral dietary supplement (120 mg highly standardized cranberry extract phytosome), compared to placebo, in diabetic postmenopausal women taking SGLT-2 inhibitors. Forty-six subjects (mean age 72.45 ± 1.76) completed the study (23 placebo/23 supplement). Considering UTI episodes, during the six-month supplementation period, an increase of 1.321 (95% CI: −0.322; 2.9650) was observed in the placebo group, while it remained at a steady value of 0.393 (95% CI: −4.230; 5.016) in the supplemented group. Regarding UTI episodes, in both groups, interaction between times for supplementation was statistically significant (*p* = 0.001). In particular, at follow-up 4, 9% of the placebo group showed infection versus only 3% with cranberry supplementation. Glycaemia and glycated hemoglobin values (secondary outcomes) were not modified at the end of six months with respect to the basal values in both groups, as expected. While in terms of quality of life per the SF-12 health questionnaire, there were no differences between the two groups, an improvement in SF-12 quality of life was observed in both groups (six months vs. basal). In conclusion, highly standardized cranberry extract phytosome supplementation reduced UTI recurrence.

## 1. Introduction

Diabetes mellitus represents an expanding global health problem. According to the data published in the 2021 Report to Parliament on diabetes mellitus, elaborated by the Ministry of Health, Italians affected by type 2 diabetes represent about 5% of the population, that is, over 3 million people. It is estimated, however, that to this percentage can be added about 1 million people who have the disease but still do not know it [1]. The estimation for global diabetes prevalence in 2045 is up to 12.2%, consisting of 783.2 million people (20–79 years old) [2].

Sodium glucose co-transporter 2 (SGLT-2) inhibitors, flozins or gliflozins, are a class of medications known for type 2 diabetes mellitus management. It has been reported that SGLT-2 inhibitors are effective in improving glycemic control and reducing hemoglobin A1c (HbA1c), body weight, systolic body pressure, and diastolic body pressure. However, patients taking this type of chronic therapy may report urinary tract infections (UTIs) [3,4]. Other clinical studies have confirmed that the use of SGLT-2 inhibitors caused UTIs of mild or moderate intensity, compared with placebo or other antidiabetic drugs [5,6]. A very recent analysis by the Food and Drug Administration (FDA) Adverse Event Reporting System, which considered a total of 45,256 reports related to the use of SGLT-2 inhibitors, including 1714 UTI cases and 438 genital mycotic infections (GMI) cases, suggested a strong association between SGLT-2 inhibitors and UTIs/GMIs, providing real-world evidence of the potential risk of UTIs/GMIs related to SGLT-2 inhibitors [7]. The most frequent infections in women (postmenopausal or not) are represented by UTI episodes [8]. Moreover, it has long been recognized that women with diabetes mellitus are possibly at a higher risk [9]. Menopause is known to predispose women to recurrent UTIs, due to lower estrogen levels and resulting changes in the urothelial and urogenital microbiome [10]. Therefore, in postmenopausal women, a relationship has been proposed between reduced abundance of vaginal commensal microbes and increased risk of urinary tract infections [11,12]. A recent study showed that consuming cranberry juice can also help vaginal microbiota health in menopausal women [13].

The fifth update of the Cochrane Review (50 studies with 8857 participants), which has been recently published, supports the use of cranberry products to reduce the risk of symptomatic, culture-verified UTIs in women with recurrent UTIs [14]. The current evidence indicates that cranberry (*Vaccinium macrocarpon*) decreases the occurrence of UTIs in elderly care residents, who are likely to benefit from the use of cranberry supplements as a preventative measure [15]. A recent study conducted in 33 women affected by recurrent uncomplicated UTIs showed that after six months of cranberry supplementation, subjects showed a significant decrease in UTIs from 2.2 ± 0.8 to 0.5 ± 0.9 (*p* < 0.001), with a significant reduction of 68% (*p* < 0.001) in the use of antibiotics during six months of cranberry intake (0.14 ± 0.35), in comparison with six months retrospectively (1.14 ± 0.71) [16].

The stable phenolic compounds proanthocyanidins (PACs) are contained in cranberry extract, and display anti-adhesion activity against *Escherichia coli* and *Candida albicans* [17,18,19]. PACs, considered the active ingredients of cranberry, were formulated as tablets, capsules, or powder and utilized in several studies [14]. Among studies that used a solid formulation, Caljouw and colleagues assessed the effectiveness of cranberry capsules in preventing UTI in vulnerable older persons aged 65 and older living in long-term care facilities; taking 500 mg cranberry capsules with 1.8% proanthocyanidins (9 mg), twice daily for 12 months, reduces the incidence of clinically defined UTI [20]. Vostalova et al. tested the whole cranberry fruit powder (proanthocyanidin content 0.56%) for preventing recurrent UTI in 182 women with two or more UTI episodes; the intake of 500 mg of cranberry fruit powder containing 2.8 mg of PACs/day for six months was associated with a reduction in incidence of recurrent UTIs [21].

The administration of antibiotics is reduced by the use of cranberry extracts, thus decreasing antibiotic resistance [17]. In vitro antibacterial activity of concentrated cranberry juice against other pathogens such as *Staphylococcus aureus*, *Pseudomonas aeruginosa*, *Klebsiella pneumoniae*, and *Proteus mirabilis* has also been demonstrated [13,14].

In consideration of the existing evidence in the literature to date, it is helpful to recommend cranberry intake to decrease the incidence of urinary tract infections, particularly in women with recurrent urinary tract infections, such as postmenopausal women taking SGLT2. Given this background, the aim of this study is to evaluate the potential benefit, assessed by urinalysis and urine culture obtained at baseline and after two, four, and six months, of a highly standardized cranberry extract oral dietary supplement formulated in phospholipids, compared to placebo, for the prevention of UTIs in diabetic postmenopausal women chronically taking SGLT-2 inhibitors.

This study is novel and original, as no studies have been published in the literature to date in which a highly standardized cranberry extract phytosome is used in women taking SLGT-2 inhibitors, a group of subjects at high risk of developing UTI. Finally, another novelty of the present study lies in the type of formulation used, which is the phytosome formulation, characterized by its phytochemical profile that maintains the full polyphenol pattern of natural cranberry juice, allowing high biological efficiency with a lower content of PACs per unitary dose, compared to other cranberry formulations.

## 2. Materials and Methods

### 2.1. Standard Protocol Approval, Registration, and Patient Consent

This study was approved by the Ethics Committee of the University of Pavia, Italy (approval number 2207/01072022) and complied with the ethical standards laid down in the 1964 Declaration of Helsinki, with written informed consent obtained from all patients entering the pre-treatment phase. This study was registered on ClinicalTrials.gov (NCT05730998).

### 2.2. Study Design and Sample Size

This was a randomized (1:1), double-blind, placebo-controlled, parallel-group, six-month clinical supplementation study. The study was conducted at the Metabolic Division of Santa Margherita Hospital, Azienda di Servizi alla Persona, Pavia, Italy. Allocation to the supplementation groups was performed via a computer-generated random-blocks randomization list, and random assignments were concealed in sealed envelopes.

Considering the study by Ledda and colleagues of 2016 as a reference study, we determined sample size according to a primary outcome defined as “decrease in urinary tract infections” by −2.4 times (+0.8 times for those treated and +3.2 times for those with placebos), and considering two balanced groups with 1:1 allocation (n1 = n2), an effect size of 80%, an alpha significance level set at 0.05, a dropout rate of 10%, and a size of (1:1), and therefore we enrolled 50 patients in total (25 patients per arm) [22].

### 2.3. Population

Diabetic postmenopausal female subjects (disease duration > 10 years and with Charlson comorbidity index >6), aged ≥70, with a body mass index (BMI) between 20 and 30 kg/m^2^, taking SGLT-2 and with at least one episode of infection in the previous year, were enrolled in the present study [23].

Subjects with severe kidney disease, moderate-to-severe hepatic failure, endocrine diseases such as thyroid disorders, psychiatric disorders, cancer (in the previous five years), or hypersensitivity to cranberry were excluded from the study. Subjects not able to take oral therapy or with artificial nutrition and subjects with inability to adhere to the study protocol were not recruited. The subjects were recruited from the Metabolic Unit of the “Santa Margherita” Institute, University of Pavia, Italy.

### 2.4. Primary and Secondary Endpoints

The primary endpoint was the prevention of recurrence of UTI episodes during a six-month supplementation. Measurements were performed on urinalysis by microscope examination and urine culture. Secondary endpoints were: quality of life, performed by the Short-Form 12-Item Health Survey (SF-12) questionnaire, consisting of a short, generic health-status measure reproducing the physical and the mental summary scores of the SF-36; safety and compliance evaluating any adverse event and any reported interaction between supplementation and the chronic use of SGLT-2 inhibitors; and any modification/alteration of the glycemic status.

### 2.5. Laboratory Tests

Laboratory tests including urinalysis and urine culture were performed to assess the primary outcome at baseline (before supplementation) and after two, four, and six months.

Urinalysis included color, specific gravity, pH, glucose, protein, red blood cell, and white blood cell. Microscopic examinations were performed under a clinical light microscope (Olympus Opto Systems India Pvt. Ltd., New Delhi, India). A positive urinary culture was defined as growth of a single urine pathogen of >10^4^ cfu/mL urine specimen. Despite known suboptimal diagnostic accuracy, urine culture and microscopy have always been considered the diagnostic ‘gold standard’ [24,25].

Fasting blood glucose levels were measured at baseline and at the end of the study (six-month supplementation) by automatic biochemical analyzer (Hitachi 747, Tokyo, Japan). Serum concentration of HbA1c was determined by a high-performance liquid chromatographic method using automatic HbA1c analyzer (Tosoh HLC-723G7, Tokyo, Japan).

### 2.6. Supplementation

All subjects consumed one capsule per day for six months, in the morning with a meal. Subjects were randomly supplemented (in a 1:1 ratio) with Anthocran™ Phytosome™ 120 mg/capsule, or matching placebo (Indena S.p.A., Milan, Italy). Anthocran™ Phytosome™ is a food-grade formulation of cranberry extract (*Vaccinium macrocarpon* Ait.) formulated with phospholipids (sunflower lecithin, Phytosome™ technology) standardized to 6–9% PACs by spectrophotometry (DMAC spectrophotometric method). Placebo and active were indistinguishable, colorless, tasteless capsules packaged and administered in an identical manner. Placebo and active capsules had the same composition except for the active substances (PACs).

Polyphenol consumption (including vegetables and fruits rich in polyphenols) was limited during the trial and at least 72 h before study. For this reason, a registered dietician performed initial dietary counselling so that subjects consumed a diet that would maintain a prudent balance of macronutrients: 25–30% of energy from fat (cholesterol < 200 mg), 55–60% of energy from carbohydrates (10% from simple carbohydrates), and 15–20% of energy from protein [19].

### 2.7. Statistical Analysis

Baseline data are presented as the mean values ± standard deviation of the mean (SD), unless otherwise indicated. The normal distribution of the variables was checked using the Shapiro–Wilk test and using Q–Q graphs. Baseline differences in demographic and clinical characteristics between the groups (placebo and intervention) were examined using independent *t* tests.

An independent *t* test was used for continuous variables to determine differences between the two treatments (intra-group). The ANCOVA test was used to define the differences between groups in UTI episodes (times). Repeated measures analysis was performed in order to assess the interaction between time (presence of infection) and treatment. Data were adjusted by age and years of diabetes.

A *p* < 0.05 value was considered significant. Statistical Package for the Social Sciences version 20 software was used to perform the statistical analysis (SPSS Inc., Chicago, IL, USA).

## 3. Results

A total of 46 postmenopausal women (mean age 72.45 ± 1.76 years old) with a BMI of 25.97 ± 1.44 kg/m^2^ completed the study (23 placebo and 23 intervention), as reported in Table 1.

The dropout rates did not significantly differ between the two groups (flow chart in Figure 1).

Figure 1 summarizes participants’ recruitment. Initially, 53 subjects were screened and 46 recruited: during the randomization, seven subjects were dismissed because of lack of informed consent. Finally, 23 subjects were allocated in each group and none was lost to follow-up.

Baseline demographic and clinical characteristics were similar in both groups (Table 1). The only datum that appeared different at the baseline was the number of medications (the cranberry-supplemented group had an average of 0.5 medication more than the placebo group), but this was clinically irrelevant.

According to the diagnostic criteria, all 46 participants were normal or overweight at baseline with glycated hemoglobin at normal level, and no significant difference in the proportion of participants between the two groups, with the exception of medications (slightly more numerous in the cranberry-supplemented group, as commented above), were present.

In addition, as showed in Table 1, the duration of the diabetes was over 10 years in both groups. The usage in terms of duration of SGLT-2 inhibitors was equal in both groups and the levels of glycated hemoglobin (%) did not differ significantly at baseline. The changes in the secondary outcomes are reported in Table 2. Intra-group analysis did not show any change between groups. Quality of life was improved in both groups, with no differences regarding supplementation. Specifically, the cranberry-supplemented group showed a clear improvement of Physical SF-12 of + 5.15 points (95% CI: 1.21; 9.09) and Mental SF-12 of +4.07 points (95% CI: 1.63; 6.51). In a similar way, the Physical and Mental SF-12 values were also improved in the placebo group after six months with respect to the baseline value.

Regarding the UTI episodes in the intervention and placebo group (Figure 2), interaction between times for treatment was statistically significant (*p* = 0.001). At follow-up 4, 9% of subjects in the placebo group reported infection versus only 3% of the cranberry-supplemented group. At follow-up 1 and 6, the same percentage of events was recorded in both groups. In total, in the placebo group, six UTI episodes were reported during the six-month administration, while for cranberry supplementation the UTI episodes reported were one in six months, with a reduction of −22.5% observed in UTIs with respect to the placebo group.

As described in Table 3, just one infection on one occasion was detected during the follow-up in the supplementation group, while six UTI infections occurred during the same follow-up period in the placebo group.

No adverse event or safety concerns were reported in either group. Furthermore, no reported interaction between supplementation and chronic use of SGLT-2 inhibitors was observed. Compliance was maintained in both groups.

## 4. Discussion

For the first time, the present study evaluated the use of 120 mg of a highly standardized cranberry extract formulated in phospholipids for six months for its benefit in preventing recurrence of UTIs in postmenopausal women with diabetes taking SGLT-2 inhibitors. The main finding of this double-blind, placebo-controlled clinical study was a statistically significant reduction of UTI episodes in the supplemented group compared to placebo administration. For UTI prophylaxis, cranberry products containing proanthocyanidins were useful to prevent adhesion of bacteria to the urothelium [26].

Furthermore, the fifth update of the Cochrane Review (50 studies with 8857 participants), which has been very recently published, supports the use of cranberry supplements to reduce the risk of symptomatic, culture-verified UTIs in women with recurrent UTIs [14]. As for dosages, some studies have used higher amounts of cranberry extract (between 400 mg and 1000 mg per day). However, the percentages of PACs contained in them were lower than in the present study, ranging from 0.56% to 1.8% [20,21,27]. This is due to the innovative formulation used in the present study. In addition to the anthocyanidins, a particularity of the dietary supplement that was administered here is the phytosome form. Highly standardized cranberry extract phytosome is a new food-grade delivery system of a highly standardized cranberry extract reproducing the natural polyphenolic profile of the fruit. This dietary supplement is standardized to contain 6–9% cranberry proanthocyanidins (PACs), representing the novelty of the present study: that amount was lower than other formulations currently developed, but allowing a high biological efficiency. Indeed, cranberry phytosome is designed and analyzed to contain not only PACs, as is the case in many other cranberry extract supplements, but also the full polyphenol pattern of natural cranberry juice [28]. Notably, this cranberry formulation was demonstrated to deliver the full polyphenolic profiles of cranberry phytoactives and PACS metabolites (e.g., valerolactone derivatives) in urine, according to a human urine pharmacokinetics study [28]. Moreover, in vitro evidence has confirmed that the formulation of this cranberry extract in phospholipids did not change the metabolism of cranberry extract by the microbiota, maintaining the formation of metabolites proven to be active against UTI pathogens [28] and leading to a comparable profile in fecal microbial catabolites [29].

An ex vivo bioassay study on this highly standardized cranberry extract phytosome demonstrated anti-adhesion activity against *Candida albicans* [28]. The results of this study are in line with previous studies that have demonstrated effectiveness of the highly standardized cranberry extract used in this study, even if not formulated in phospholipids, on UTI episodes, although in different populations, such as young healthy subjects with recurrent UTIs [30,31] and elderly men suffering from moderate prostatic hyperplasia [22]. Recently, a preliminary pilot study demonstrated that cranberry phytosome may prove effective as supplementary management in preventing post-operative, post-catheter UTIs in subjects with or without recurrent UTIs [32].

Notably, cranberry extracts have been demonstrated to exert remarkable anti-inflammatory and antioxidant effects on transitional cells, consequently acting on residual inflammation present after UTI infections [31,32,33,34]. In this context, our cranberry formulated in phospholipids may potentially have reduced this lower residual inflammation with respect to the placebo group.

As expected, the glycemic panel was not changed by supplementation, but a significant improvement in quality of life was reported in both groups. This may be ascribed to the placebo effect that may occur in these clinical trials [35].

Despite the significant findings obtained in the present investigation, there are some limitations, such as the limited number of enrolled subjects. No differences were observed in quality of life. This may be ascribed to the limited number of subjects included in the study. Thus, future studies using larger sample sizes and including a wide spectrum of biomarkers are required. In particular, it should be noted that we did not measure antioxidant and inflammatory markers related to UTIs. In addition, our sampling was based only on diabetic postmenopausal women. Another interesting future prospective might be the experimental comparison with other cranberry formulations.

Therefore, although these preliminary results are promising, underlining also the safety and tolerability of a long-term supplementation with this cranberry formulation in phospholipids, they need further confirmation from larger cohort studies in populations other than diabetic postmenopausal women taking SGLT-2 inhibitors, a population generally at high risk of UTIs.

## 5. Conclusions

Daily administration of 120 mg of highly standardized cranberry extract formulated in phospholipids (containing 6–9% PACs), in diabetic postmenopausal women taking SGLT-2 inhibitors, led to a statistically significant reduction of UTI episodes in the supplemented group in comparison with placebo administration.

## Figures and Tables

**Figure 1 nutrients-16-02113-f001:**
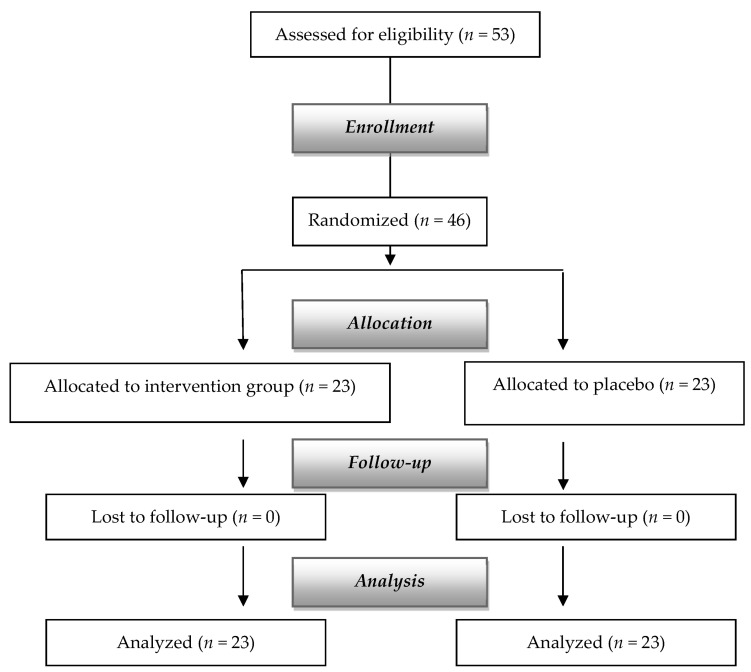
Flow diagram of the study.

**Figure 2 nutrients-16-02113-f002:**
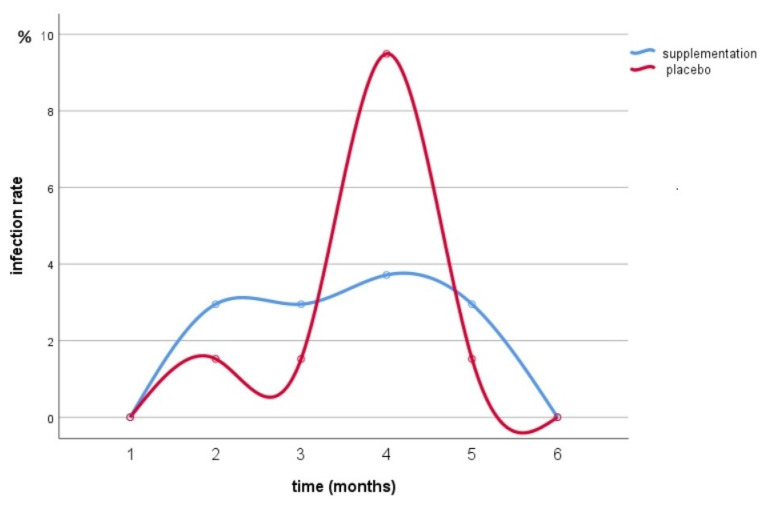
UTI episodes during the six-month supplementation.

**Table 1 nutrients-16-02113-t001:** Descriptive statistics of the sample at baseline.

Variable	A Placebo *n* = 23 (Mean ± SD)	B Supplementation *n* = 23 (Mean ± SD)	Total Sample *n* = 46 (Mean ± SD)	*p*-Value between Groups at Baseline
Age (years)	72.00 ± 1.944	72.65 ± 1.695	72.45 ± 1.769	0.338
Duration of diabetes (years)	11.50 ± 1.080	11.78 ± 0.951	11.70 ± 0.984	0.457
Use of SGLT-2 inhibitors (years)	5.40 ± 0.966	5.17 ± 1.114	5.24 ± 1.062	0.582
Glycemia (mg/dL)	116.50 ± 5.290	120.57 ± 4.771	118.34 ± 5.189	0.072
Glycated hemoglobin (%)	6.100 ± 0.082	6.165 ± 0.130	6.145 ± 0.120	0.155
Medications (number)	2.50 ± 0.707	3.35 ± 0.935	3.09 ± 0.947	**0.016**
BMI (kg/m^2^)	25.90 ± 1.524	26.00 ± 1.446	25.97 ± 1.447	0.859
Physical SF-12	36.39 ±5.26	34.54 ± 10.67	35.46 ± 8.37	0.460
Mental SF-12	41.88 ± 8.57	39.30 ± 12.34	40.59 ± 19.53	0.410

Abbreviations: BMI, body mass index. In bold value with *p* < 0.05.

**Table 2 nutrients-16-02113-t002:** Within-group mean changes from baseline (from day 0) to the end of the supplementation (six months) for all investigated variables related to secondary endpoints.

Variable	Intra-Group Δ Change (CI 95%)	*p*-Value
A. placebo		
Glycemia (mg/dL)	−1.739 (−3.810; 0.332)	0.096
Glycated hemoglobin (%)	−0.017 (−0.074; 0.039)	0.528
Physical SF-12	2.072 (0.311; 3.832)	**0.023**
Mental SF-12	0.995 (0.330; 1.661)	**0.005**
B. supplementation		
Glycaemia (mg/dL)	0.261 (−2.261; 2.783)	0.832
Glycated haemoglobin (%)	0.000 (−0.058; 0.058)	1.000
Physical SF-12	5.156 (1.216; 9.096)	**0.013**
Mental SF-12	4.078 (1.639; 6.516)	**0.002**

In bold value with *p* < 0.05.

**Table 3 nutrients-16-02113-t003:** Number of subjects reporting UTI episodes during six-month supplementation.

		Number of Infections during 6 Months				
					Total UTI Episodes	
	Total Subjects	0 Times	1 Time	2 Times	Number	%
Placebo	23	18	4	1	6	26
Supplementation	23	22	1	0	1	4
Total	46	40	5	1	7	15

## Data Availability

The raw data supporting the conclusions of this article will be made available by the authors on request.

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
