# Peer review of "Supplementation with Highly Standardized Cranberry Extract Phytosome Achieved the Modulation of Urinary Tract Infection Episodes in Diabetic Postmenopausal Women Taking SGLT-2 Inhibitors: A RCT Study"

_nutrients, 2024, doi:10.3390/nu16132113_

Round 1

Reviewer 1 Report

Comments and Suggestions for Authors

This research is important and brings valuable information with practical application that uses an appropriate experimental design. The research presented is well planned and the manuscript is well organized. The authors use relevant, explicit iconography. Therefore, the paper might be of interest, but some points should be reconsidered before acceptance:

The title of the article is too long, with too many experimental details. A shorter, more concise reformulation of the title would be necessary.

Lines 28-29 - the wording is not clear, what do the numerical values refer to?

In Introduction. This part lacks a comparison with similar studies performed in scientific literature. Although some studies are presented (their presentation needs improvement), no sufficient comparisons with the purposes of your study are found. This will help to highlight the novelty and originality of the present work. Novelty and originality should afterwards clearly be stated in order to highlight what your study brings in novelty in the field of research, compared to other studies with similar subjects. The last paragraph needs serious improvement, as it does not explain properly the purpose and neither novelty nor originality. All these should be highlighted in order to increase the value of the results obtained.

The names of the figures must be placed under the figures.

More discussions related to previous similar studies carried out on cranberry extract (even in other forms of administration) in the treatment of urinary infections would be necessary.

Due to the limitations of the current research, the experimental design could have included patients to whom the cranberry extract contains proanthocyanidins in a different administration form than phytosomes. This would allow a series of discussions related to the influence of the form of administration on the bioavailability of the active components.

Other parameters that should be taken into account are those related to the bioavailability of proanthocyanidins in patients.

More bibliographic references would be needed for these discussions.

A chapter of Conclusions, with the most important results obtained in the present study and the perspectives of future research to cover the limitations of this study would be necessary.

Author Response

Dear Reviewer,

Thank you for your suggestions.

The manuscript has been modified as requested. The changes in the text are highlighted in yellow.

Moreover, the degree of overlap has been reduced, following the report provided to us.

Best regards,

The authors

Reviewer 1

This research is important and brings valuable information with practical application that uses an appropriate experimental design. The research presented is well planned and the manuscript is well organized. The authors use relevant, explicit iconography. Therefore, the paper might be of interest, but some points should be reconsidered before acceptance:

The title of the article is too long, with too many experimental details. A shorter, more concise reformulation of the title would be necessary.

Answer: The title has been shortened, accordingly, as highlighted in the manuscript’s version with track changes.

Lines 28-29 - the wording is not clear, what do the numerical values refer to?

Answer: The sentence has been rearranged to better explain all the numerical values.

In Introduction. This part lacks a comparison with similar studies performed in scientific literature. Although some studies are presented (their presentation needs improvement), no sufficient comparisons with the purposes of your study are found. This will help to highlight the novelty and originality of the present work. Novelty and originality should afterwards clearly be stated in order to highlight what your study brings in novelty in the field of research, compared to other studies with similar subjects. The last paragraph needs serious improvement, as it does not explain properly the purpose and neither novelty nor originality. All these should be highlighted in order to increase the value of the results obtained.

Answer: Thank you for your kind suggestions. In the introduction further studies and therefore their references have been included and commented. Considering the innovation and originality of the study, an explanatory sentence has been added, accordingly.

The names of the figures must be placed under the figures.

Answer: The figure captions have been moved under the figures.

More discussions related to previous similar studies carried out on cranberry extract (even in other forms of administration) in the treatment of urinary infections would be necessary.

Answer: The discussion has been rearranged and improved as suggested.

Due to the limitations of the current research, the experimental design could have included patients to whom the cranberry extract contains proanthocyanidins in a different administration form than phytosomes. This would allow a series of discussions related to the influence of the form of administration on the bioavailability of the active components.

Other parameters that should be taken into account are those related to the bioavailability of proanthocyanidins in patients.

More bibliographic references would be needed for these discussions.

Answer: Thank you for your comments and suggestions. Other studies that used different cranberry’s forms and the related references have been now included in the discussion. As regards the comment "the experimental design could have included patients to whom the cranberry extract contains proanthocyanidins in a different administration form than phytosomes" this suggestion has been included as future prospective.

A chapter of Conclusions, with the most important results obtained in the present study and the perspectives of future research to cover the limitations of this study would be necessary.

Answer: A chapter of Conclusions has been now added, accordingly.

Reviewer 2 Report

Comments and Suggestions for Authors

Title: Six-months daily supplementation with 120 mg of highly standardized cranberry extract Phytosome achieved the modulation of urinary tract infection (UTI) episodes in diabetic post-menopausal women taking sodium glucose co-transporter 2 (SGLT-2) inhibitors: a randomized, double-blind, placebo-con-trolled trial.

This study promotes the utilization of the use of a nutritional supplement to reduce cases of urinary infections in high-risk populations such as menopausal women with type 2 diabetes. However, the manuscript presents a series of drawbacks that need to be corrected before the manuscript can be published. In the following lines I will explain the main mistakes found.

Line 47-48. Is this still type 2 diabetes data?

Line 56. Rewrite.

Line 57. Clarify the meaning of the acronym FDA.

Line 57-61. Rewrite. Change “was observed”. Furthermore, the sentence is too complex. Authors should consider reducing sentence length to increase text compression.

Line 62-63. This information is repeated from the abstract. Furthermore, it would be convenient to relocate this information elsewhere. It is also not appropriate to make paragraphs of just a few lines.

Line 73-74. Again, a paragraph of only two lines. There cannot be paragraphs of this type. Furthermore, this information is disordered. It happens several times throughout the text. It is necessary to reorganize the information.

Line 81. Delete “very”.

Line 83-88. Again, disorganized information. Text cohesion needs to be improved.

Line 122. Rewrite. First person verbs should not be used in a scientific article.

Line 127. Diabetes is an endocrine disease. It cannot be a exclude conditions. You should put other endocrine diseases.

Line 136. Delete “:”.

Line 172. The consumption of vegetables and fruit is recommended in diabetic patients. What kind of diet did they follow during the study? Protein diet?

Line 192. In line 118, it was mentioned that the study was going to be carried out with 50 patients. In the rest of the article, you put 46. What happened to those 4 that are missing?

Figure 1. Delete. The results section must be significantly improved. The authors practically limit themselves to putting the tables with the numerical results and it is the reader who has to stop and interpret the data.

Line 259-267. The text keeps repeating the same information all the time. It seems that the supplement company paid for the study to promote its potential benefits.

In addition, these types of supplements have been available on the market for a long time in order to prevent infections and their consumption is recommended by many specialists. So, what new does this study provide? That other patent also seems to reduce cases of infections?

Line 274. “In vitro” should be in italics. All Latin expressions must be italicized. Review the entire manuscript since this error is repeated in more sections of the text.

Line 288. Has anything been carried out in this line of research in this study to be able to compare in the discussion? Since it has not been carried out, it would be more appropriate to put this information in the introduction than in a discussion section. The authors limit themselves to saying that it is done in the bibliography.

Comments on the Quality of English Language

Moderate editing of English language required

Author Response

Dear Reviewer,

Thank you for your suggestions.

The manuscript has been modified as requested. The changes in the text are highlighted in yellow.

Moreover, the degree of overlap has been reduced, following the report provided to us.

Best regards,

The authors

Reviewer 2

This study promotes the utilization of the use of a nutritional supplement to reduce cases of urinary infections in high-risk populations such as menopausal women with type 2 diabetes. However, the manuscript presents a series of drawbacks that need to be corrected before the manuscript can be published. In the following lines I will explain the main mistakes found.

Line 47-48. Is this still type 2 diabetes data?

 Answer: Some data reported are specific to type 2 diabetes as specified; references 2 instead deals with diabetes mellitus not otherwise specified. For this reason the first sentence has been slightly modified.

Line 56. Rewrite.

Answer: the sentence has been rearranged.

Line 57. Clarify the meaning of the acronym FDA.

Answer: the acronym has been clarified.

Line 57-61. Rewrite. Change “was observed”. Furthermore, the sentence is too complex. Authors should consider reducing sentence length to increase text compression.

Answer: the sentence has been modified.

Line 62-63. This information is repeated from the abstract. Furthermore, it would be convenient to relocate this information elsewhere. It is also not appropriate to make paragraphs of just a few lines.

 Answer: The sentences have been removed and the paragraph revised, accordingly.

Line 73-74. Again, a paragraph of only two lines. There cannot be paragraphs of this type. Furthermore, this information is disordered. It happens several times throughout the text. It is necessary to reorganize the information.

 Answer: Sentence have been rearranged.

Line 81. Delete “very”.

 Answer: deleted

Line 83-88. Again, disorganized information. Text cohesion needs to be improved.

Answer: Sentence have been revised and rearranged

Line 122. Rewrite. First person verbs should not be used in a scientific article.

 Answer: The sentence has been re-written.

Line 127. Diabetes is an endocrine disease. It cannot be a exclude conditions. You should put other endocrine diseases.

 Answer: The sentence has been clarified.

Line 136. Delete “:”.

 Answer: deleted. The sentence has been modified.

Line 172. The consumption of vegetables and fruit is recommended in diabetic patients. What kind of diet did they follow during the study? Protein diet?

 Answer: A sentence has been added on this topic

Line 192. In line 118, it was mentioned that the study was going to be carried out with 50 patients. In the rest of the article, you put 46. What happened to those 4 that are missing?

 Answer: The number 50 represents the reference sample size to aim for. In our study 53 subjects were eligible and screened, of these 46 were subsequently enrolled, as shown in the flowchart (figure 1)

Figure 1. Delete. The results section must be significantly improved. The authors practically limit themselves to putting the tables with the numerical results and it is the reader who has to stop and interpret the data.

 Answer: We believe the flowchart (figure 1) is useful, as for general MDPI guidelines for authors; the results section has been improved as suggested.

Line 259-267. The text keeps repeating the same information all the time. It seems that the supplement company paid for the study to promote its potential benefits.

 Answer:  Anthocran phytosome utilized in the present Study was produced by Indena S.p.A. as written in the paper. However, the clinical study was performed according the ethical standards of Declaration of Helsinki and the Good Clinical Practice procedures and Indena did not participate and did not influence in any way the outcome of the Study

In addition, these types of supplements have been available on the market for a long time in order to prevent infections and their consumption is recommended by many specialists. So, what new does this study provide? That other patent also seems to reduce cases of infections?

 Answer: Novelty and originality have been described in the introduction, as required.

Line 274. “In vitro” should be in italics. All Latin expressions must be italicized. Review the entire manuscript since this error is repeated in more sections of the text.

 Answer: Italics has been used along the text.

Line 288. Has anything been carried out in this line of research in this study to be able to compare in the discussion? Since it has not been carried out, it would be more appropriate to put this information in the introduction than in a discussion section. The authors limit themselves to saying that it is done in the bibliography.

Answer: More information has been included in the introduction and new references have been added in the discussion, as suggested.

Round 2

Reviewer 1 Report

Comments and Suggestions for Authors

The changes made by the authors bring value to the research, complete the scientific information with clarifying data. Thus, the paper overlaps the requirements of the journal.

Author Response

Thank you so much for appreciating revisions made.